# From Environment to Gene Expression: Epigenetic Methylations and One-Carbon Metabolism in Amyotrophic Lateral Sclerosis

**DOI:** 10.3390/cells13110967

**Published:** 2024-06-03

**Authors:** Marina Hernan-Godoy, Caroline Rouaux

**Affiliations:** Inserm UMR_S 1329, Strasbourg Translational Neuroscience and Psychiatry, Université de Strasbourg, Centre de Recherche en Biomédecine de Strasbourg, 1 Rue Eugène Boeckel, 67 000 Strasbourg, France; hernangodoy@unistra.fr

**Keywords:** amyotrophic lateral sclerosis, one-carbon metabolism, epigenetics, DNA methylation, histone methylation, therapeutic strategy

## Abstract

The etiology of the neurodegenerative disease amyotrophic lateral sclerosis (ALS) is complex and considered multifactorial. The majority of ALS cases are sporadic, but familial cases also exist. Estimates of heritability range from 8% to 61%, indicating that additional factors beyond genetics likely contribute to ALS. Numerous environmental factors are considered, which may add up and synergize throughout an individual’s lifetime building its unique exposome. One level of integration between genetic and environmental factors is epigenetics, which results in alterations in gene expression without modification of the genome sequence. Methylation reactions, targeting DNA or histones, represent a large proportion of epigenetic regulations and strongly depend on the availability of methyl donors provided by the ubiquitous one-carbon (1C) metabolism. Thus, understanding the interplay between exposome, 1C metabolism, and epigenetic modifications will likely contribute to elucidating the mechanisms underlying altered gene expression related to ALS and to developing targeted therapeutic interventions. Here, we review evidence for 1C metabolism alterations and epigenetic methylation dysregulations in ALS, with a focus on the impairments reported in neural tissues, and discuss these environmentally driven mechanisms as the consequences of cumulative exposome or late environmental hits, but also as the possible result of early developmental defects.

## 1. Introduction

The fatal neurodegenerative disease amyotrophic lateral sclerosis (ALS) arises from the combined and progressive loss of neuronal populations involved in voluntary movements and present in the motor cortex, brain stem, and spinal cord: the corticospinal neurons (CSNs, or upper motor neurons) and the motoneurons (MNs, or lower motor neurons), respectively [1]. While ALS is highly variable in its initial presentation, age of onset, and duration, it is generally defined as an adult-onset disease that leads to progressive paralysis and death within only 2 to 3 years following first motor symptom appearance. Histologically, ALS is also characterized by the intracytoplasmic phospho-TDP-43 protein inclusions, the so-called TDP-43 pathology [2,3].

ALS etiology is complex and now considered multifactorial. Genetics plays a significant role, as ALS can be inherited, with familial cases (fALS) making up 15% of instances while the remaining 85% are sporadic (sALS). Approximately 70% of fALS patients carry an identified ALS-associated genetic mutation, compared to only 15% of sporadic cases. Most of the genetic cases identified so far arise from autosomal dominant mutations in more than 20 genes, the 5 most frequent being *C9ORF72* (hereafter, shortened as *C9*), *SOD1*, *FUS*, *TARDBP*, and *TBK1* [1]. Estimates of heritability, which vary between populations, range from 8% to 61%, indicating that additional factors beyond genetics likely contribute to ALS [1] (Figure 1A).

If genetic susceptibility is by nature inherited and remains identical throughout the entire life duration, environmental contribution to health and disease is constantly evolving. On this basis, three models of interactions between genetics and environment can be envisioned that would trigger disease onset. The first model is the result of interaction between genomic susceptibility and the exposome [4], in which environmental factors add up and synergize throughout an individual’s lifetime. This model is in accordance with the gene–time–environment hypothesis [5]. The second model is the result of interaction between genomic susceptibility and a detrimental environmental event occurring soon before and triggering disease onset, in accordance with the multistep hypothesis [6]. The third model is the result of interaction between genomic susceptibility and a detrimental environmental event occurring during development, long before disease onset, and which immediate consequences would remain silent or compensated for decades, in accordance with the developmental hypothesis of neurodegenerative diseases [7,8] (Figure 1B).

One level of integration between genetic and environmental factors is epigenetics [9], which is altogether impacted by environmental factors, either directly or through the metabolome, and results in alterations of gene expression without modification of the genome sequence. If epigenetic alterations nicely fit within all of the above-mentioned hypotheses, they represent a preferred mechanism for retaining the memory of developmental events into adulthood. Three main types of epigenetic regulations are described: DNA methylation, histone post-translational modifications, which include methylation and acetylation reactions, and miRNA expression.

Post-translational histone acetylation relies on the activities of two major classes of enzymes, namely histone acetyltransferases (HATs) and histone deacetylases (HDACs), and has been studied in particular for its impact on gene expression [10] and in various neuropathological contexts, including ALS [11]. In particular, HDAC inhibitors have been tested for their ability to counteract disease progression in mouse models of the disease [12,13,14,15]. The pan-HDAC inhibitor sodium phenylbutyrate has been tested in clinical trials in combination with taurursodiol (AMX0035) [16], which unfortunately failed in a phase III trial. However, it should be noted that the effects of HDAC inhibitors go beyond histone acetylation status alone and that their consequences on gene expression are not routinely assessed [12,17,18,19].

miRNAs are small, non-coding, single-stranded RNA of about 18–25 nucleotides involved in the regulation of post-transcriptional gene expression acting on the stability and translation rate of mRNAs [20]. In addition, evidence indicates that transcription factors as well as epigenetic regulators such as DNA methyltransferases and histone-modifying enzymes are highly probable miRNA targets, providing additional indirect mechanisms for miRNA contribution to epigenetic regulations of gene expression [20]. The implication of miRNA in the pathophysiology of ALS, in particular, in relation to the global RNA metabolism alterations that characterize the disease and the impairment of RNA-binding proteins (RBPs) such as FUS and TDP-43, and their potential as biomarkers and therapeutic targets have recently been reviewed elsewhere [21,22].

Methylation reactions targeting either DNA or histones represent a large proportion of epigenetic regulations. If DNA methylation is mostly associated with gene silencing, histone methylation is more versatile depending on the targeted residue and the number of methyl groups (detailed below). Methylation reactions strongly depend on the availability of methyl donors, which are provided by the ubiquitous one-carbon (1C) metabolism pathway [23]. Thus, understanding the interplay between exposome, 1C metabolism, and epigenetic modifications will likely contribute to elucidating the mechanisms underlying altered gene expression related to neurodegeneration in general and ALS in particular and to developing targeted therapeutic interventions (Figure 1C).

Here, we review evidence for 1C metabolism alterations and methylation-related epigenetic dysregulations in ALS. Because epigenetic regulations are highly tissue- and cell-type specific [24], we mostly focused this review on neural tissues where epigenetic and metabolomic modulations are more likely to reflect neurodegenerative processes. We finally discuss these environmentally driven mechanisms as the consequences of cumulative exposome or late environmental hits, but also as the possible result of early developmental defects.

## 2. One-Carbon Metabolism

One-carbon metabolism is a central and ubiquitous cellular pathway composed of two intertwined cycles: the folate cycle whose main output is purine synthesis, necessary for DNA replication during proliferation, and the methionine cycle whose main outputs are methyl donors used for methylation reactions, including chromatin methylations [23]. Key enzymes of the pathway are (1) dihydrofolate reductase (DHFR), upstream of both cycles and responsible for converting dihydrofolate (DHF) into tetrahydrofolate (THF); (2) methylenetetrahydrofolate reductase (MTHFR), which generates 5-methyl-THF (5-MTHF) from 5,10-methylene-THF (5,10-MTHF); (3) S-Adenosylmethionine synthase (MAT2A), which metabolizes methionine into the universal methyl donor S-Adenosylmethionine (SAM); and (4) adenosylhomocysteinase (ACHY), which converts the compound S-adenosylhomocysteine (SAH) to adenosine and homocysteine (HCY) (Figure 2).

### 2.1. One-Carbon Metabolism in ALS

Alterations of 1C metabolism, and more particularly of the folate and methionine cycles, have been related to neurological disorders and neurodegeneration [25] including ALS (reviewed in [26]). Several reports revealed polymorphisms in the gene encoding MTHFR that catalyses the re-methylation of HCY to methionine [27,28]. Other polymorphisms of genes encoding the paraoxonases (PON 1, 2, 3) that convert HCY to homocysteic acid have also been reported [29]. In comparison with controls, ALS patients present decreased folate levels in plasma [30] and increased levels of HCY in cerebrospinal fluid, plasma and serum [26,30,31,32] (Figure 2). Importantly, HCY levels in ALS seem positively correlated with the rate of disease progression [30]. Like in patients, decreased levels of folic acid were reported in the plasma, cerebral cortex, and spinal cord of middle- to late-stage *SOD1^G93A^* mouse model of ALS, along with reduced levels of 5-MTHF at earlier time points [33]. One-carbon metabolism and, more particularly, folate cycle impairment in *SOD1^G93A^* mice were recently confirmed by a metabolomic study conducted on spinal cord samples [34]. The relevance of 1C metabolism impairment in ALS was further demonstrated in two studies, which reported the beneficial effects of either folic acid or SAM supplementation in *SOD1^G93A^* mice [35,36] (Figure 2).

HCY is a non-protein thiol-containing amino acid, precursor of methionine, and its metabolism is mainly dependent on vitamin cofactors such as folate and vitamin B12. Thus, low folate and/or B12 diet could account for increased HCY levels [30]. While the origins of increased HCY in the context of ALS still remain to be fully explained, their consequences on the central nervous system (CNS) instead are better understood. HCY is neurotoxic, both per se and via its conversion into homocysteic acid (HCA) and homocysteine thiolactone. Both HCY and HCA can interact with NMDA and non-NMDA glutamatergic receptor subtypes leading to excitotoxicity and neurodegeneration of cortical neurons in vitro and motor neurons in vivo [37,38]. In addition, HCA drives hyperexcitability and seizures in vivo [39]. Homocysteine thiolactone, a neurotoxic reactive thioester, can also lead to neuronal hyperexcitability by inducing a strong reduction in the Na^+^/K^+^ ATPase activity in various structures of the CNS [40]. Finally, increased urine levels of methylmalonate (MMA), another derivative of HCY produced by cystathionine synthase, have been reported in ALS patients [31], a phenomenon demonstrated to be neurotoxic [41] (Figure 2). 

### 2.2. One-Carbon Metabolism and Epigenetics

The link between 1C metabolism and epigenetics has been reported in several physiological and pathological contexts such as development, cancer, and neurodegeneration [42,43,44]. In addition to providing methyl residues necessary for chromatin methylation reactions, metabolites and enzymes of 1C metabolism contribute to finer epigenetic regulations. For instance, both SAH and high concentrations of HCY negatively impact methyltransferase activity (reviewed in [45]), while MTHFR participates in the maintenance of heterochromatin, which corresponds to a highly condensed region of transcriptionally silent chromatin [46]. Hyperhomocysteinemia, which is accompanied by a decreased SAM/SAH ratio as reported in several organs including the brain, liver, heart, and kidney, also associates with either increased or decreased levels of DNA and histone methylation (reviewed in [47]). Therefore, 1C metabolism contributes to epigenetic modifications and by extension to the regulation of gene expression by different means that appear to be more diversified than initially thought.

## 3. DNA Methylation

The first layer of epigenetic regulation of the genome is DNA methylation (DNAm). It consists in the covalent addition of a methyl group on a cytosine nucleotide leading to 5-methylcytosine (5mc) formation [48]. The 5mC are mostly found in CpG islands, which correspond to genome sequences enriched in cytosine/guanine dinucleotides (CpG) mostly located in the promoter regions of genes. DNAm of CpG islands modifies the structure of the chromatin within promoter regions, preventing the binding of transcription factors and consequently leading to gene silencing in a dose-dependent manner [48]. DNAm results from the activity of two classes of DNA methyltransferases (DNMTs). DNMT1 maintains heritable DNAm during cell replication, targeting solely hemi-methylated residues, whereas DNMT3A, DNMT3B, and DNMT3L catalyze de novo methylation, targeting either non-methylated residues or hemi-methylated residues [48]. Noteworthily, DNMT3A subcellular localization is not only nuclear, but has also been identified in the mitochondria where it may methylate mitochondrial DNA as well as in presynaptic MN terminals where its substrates and functions have not been identified yet [49]. If DNAm is a one-step process, demethylation instead requires multiple steps and involves several enzymatic players that are not all fully described, starting with an oxidation of the methyl residue by ten-eleven translocation enzymes (TETs) and leading to the formation of 5-hydroxymethylcytosine (5hmC, reviewed in [48,50]). While this contributes to view DNAm as the most stable and inheritable epigenetic mark, it should, however, be emphasized that DNAm is extremely dynamic with fast modulations occurring within the time frame of the circadian rhythm and slower modulations towards either hyper- or hypomethylation of CpG islands appearing over the entire lifespan (reviewed in [51]). These long-term dynamics permitted the emergence of the so-called DNAm clocks, which can acutely assess the chronological age of individuals, and reflects the combination of genetics, epigenetics, and environmental factors within their biological age [52]. In all, DNAm is a fundamental gene expression regulatory element, associated with physiological and pathological developmental processes as well as physiological and pathological aging [48,50,52].

### 3.1. Increased Expression of DNMTs and DNAm in ALS Is Associated with Neuronal Death

Mutations in *DNMTs* are known to cause neurodevelopmental and neurodegenerative disorders. De novo germline pathogenic mutations of *DNMT3A* have been associated with the Tatton–Brown–Rahman syndrome, characterized by overgrowth, macrocephaly, facial features, and mild-to-severe intellectual disability [53,54], and *DNMT1* mutations with the neurodegenerative hereditary sensory and autonomic neuropathy type 1 and autosomal dominant cerebellar ataxia [55,56]. As of today, no mutation or rare variants of *DNMTs* have been reported in ALS. However, altered *DNMT* expression has been described in the tissues of ALS patients, as well as in cell cultures and mouse models of the disease, and is summarized in Table 1. In sALS patients, DNMT1 and DNMT3A protein levels were found increased in the pyramidal neurons of the motor cortex and MN of sALS patients [49], together with 5mC accumulation in the nuclei of these same neurons and of parvalbumin-positive spinal interneurons [49,57]. Appleby-Mallinder et al. confirmed increased 5mC in MN from sALS and *C9*-ALS patients, as well as increased 5hmC levels [58]. Interestingly, the authors also unraveled a relationship between DNAm and TDP-43 proteinopathy by evidencing that nuclear depletion of TDP-43 was accompanied by lower levels of both 5mC and 5hmC [58]; however, the cause-and-effect relationship between these two cellular events has yet to be demonstrated. Similarly, in *FUS*-ALS patient-induced MN, expression of *DNMT1* and *DNMT3A* was found greatly enriched compared to controls [59]. Finally, in the *SOD1^G93A^* and *SOD1^G37R^* mouse models of ALS, DNMT1 and DNMT3A protein levels and binding to chromatin as well as global DNMT enzymatic activity were reported increased both in spinal cord and skeletal muscle and were accompanied by increased 5mC levels in MN and satellite cells of the skeletal muscle [57]. Overall, these tissue- and model-based studies point to a global increase in DNMT and DNAm in ALS. It is, however, noteworthy that a recent single-nucleus RNA sequencing (snRNA-seq) database generated from post-mortem motor and frontal cortices from *C9*-ALS and *C9*-frontotemporal dementia (FTD) patients along with control donors reported decreased DNMT3A expression in upper and lower cortical neurons [60]. Epigenetic alterations were investigated in parallel by the means of single-nucleus sequencing assay for transposase-accessible chromatin (snATAC-seq) and chromatin immunoprecipitation (ChipSeq) [60], but global DNAm was not assessed, preventing comparisons with former studies.

A first causal link between DNMT protein levels and activity, DNAm, and neurodegeneration was provided by Chestnut and collaborators. Using cellular and animal models, they demonstrated that neuronal death is accompanied by increased DNMT1 and DNMT3A protein levels and activities, which reflect in nuclear accumulation of 5mC. Importantly, overexpression of DNMT3 was sufficient to trigger neuronal death of cultured NSC34 cells, while blocking DNMT activity with RG108 or procainamide, both in the dish and in rodent models of MN degeneration, was sufficient to prevent 5mC accumulation and neuronal apoptosis [49]. In the *SOD1^G37R^* mouse model of ALS, chronic and systemic RG108 administration delayed motor onset and improved motor function and survival [57]. Together, these studies indicate that DNMT expression and activity, as well as global DNAm, are found increased in affected neural tissues from ALS patients and cellular and animal models of the disease, in which they likely reflect ongoing neurodegeneration. Interestingly, this pattern is reminiscent of those reported in other neurodegenerative diseases, such as Alzheimer’s disease (AD), Parkinson’s disease (PD), or Huntington’s disease (HD) [61].

### 3.2. DNAm Age Contributes to ALS Onset and Heterogeneity

Age is the first risk factor of sALS [62]. Age also reflects in the methylation status of our genome [63]. Numerous epigenetic clocks have been developed [64], in particular, the Horvath’s clock, which serves as a highly reliable indicator of chronological age across various tissues [63]. This metric evaluates the methylation status of 353 CpGs situated within genes pivotal to cellular processes such as cell death/survival, cellular growth/proliferation, organism/tissue development, and cancer [63]. Among the 353 CpGs, 193 get hypermethylated and 160 get hypomethylated with age [63]. DNAm age acceleration is calculated as the difference between DNAm age and chronological age. Studies have consistently revealed DNAm age acceleration in neurodegenerative disorders including AD, PD, and HD, but also Down syndrome, among other pathological conditions (reviewed in [64]). 

Global DNAm was first assessed on blood samples from ALS patients (mostly sporadic) and controls and demonstrated a 25 to 30% increase in patients compared to controls [65]. The team of Ekaterina Rogaeva was the first to address the question of DNAm age acceleration in the context of ALS [66,67]. Using blood samples and post-mortem CNS samples from *C9*-ALS patients [67] or sporadic patients [66], the team calculated DNAm age and DNAm age acceleration. It is important to highlight that, independently of the tested tissue, ALS does not associate with an increased DNAm age, and DNAm age acceleration spans from negative to positive values [66,67]. This is in accordance with the first epigenome-wide association study (EWAS) conducted on post-mortem frontal cortex from sALS patients and control individuals that revealed no correlation between DNAm levels and ages of the individuals, whether they were sALS patients or controls [68]. Yet, DNAm age acceleration correlates with age at onset both in sALS and *C9*-ALS patients [66,67], as well as disease duration in *C9*-ALS patients [67] and survival in sALS patients [66], suggesting that DNAm age acceleration may contribute to the high heterogeneity of ALS.

In this context, the case of monozygotic twins, in which the disease was demonstrated to be discordant in over 90% of the cases [69], is particularly informative. DNAm age was repeatedly reported to be higher in individuals with ALS in comparison with their monozygotic twin, or triplets [70,71,72], not only in sALS cases [70,71], but also in *SOD1*-ALS [71,72] and *C9*-ALS cases [71], indicating that DNAm might strongly influence the penetrance of a causative mutation.

Together, these studies highlight not only DNAm age acceleration as a putative useful biomarker in disease onset, progression, and survival, but also suggest that it might represent a potentially interesting therapeutic target, proven that it can be manipulated in a cell-type and disease-specific manner.

### 3.3. DNAm Changes Confirm Transcriptomic Analyses

The EWAS conducted by Morahan and colleagues in 2009 was the first study to investigate the role of DNAm in ALS [68]. The team assessed DNAm in the post-mortem frontal cortex of male sALS patients and control individuals using ChIP-on-Chip [68]. Gene ontology (GO) analysis highlighted calcium dynamics, excitotoxicity, oxidative stress, neuronal exocytosis, and brain development [68]. Within these categories, genes were either hyper- or hypomethylated in sALS patients compared to controls, and these differences were present either on identified CpG islands or within the gene domains (exons, introns, and flanking areas). A second EWAS study by Figueroa-Romero and colleagues assessed the levels of 5mC and 5hmC in the spinal cord of sALS patients versus control individuals using the DNA bisulfite conversion method. The authors confirmed a 1.4-fold increase in DNAm in the tissue from ALS patients compared to controls, but also a 3-fold increase in the 5hmC mark [73]. The work further combined DNAm analysis with gene expression analysis, assessed by microarray, compared differentially methylated genes and differentially expressed genes to unravel “concordant epigenes” that showed concordant methylation and expression status, and highlighted mostly biological responses associated with inflammation and immune response [73]. Appleby-Mallinder et al. (2021) conducted a methylation study on post-mortem MN purified from sALS and *C9-*ALS patients [58]. They identified 732 differentially methylated promoters, of which 330 were hypomethylated by 9.4% on average and 402 were hypermethylated by 21.8% on average. GO analysis revealed that hypomethylated genes were associated with urea metabolism, while hypermethylated genes were associated with RNA metabolism and splicing [58], in accordance with genetic and transcriptomic data [74].

In their 2017 study, Ebbert and collaborators integrated and compared the methylome and transcriptome profiles of post-mortem frontal cortex and cerebellum samples from *C9*-ALS, sALS, and controls. They showed “distinct but overlapping” methylome profiles between *C9*-ALS and sALS [75]. In particular, they identified the demethylation and de-repression of *SERPINA1*, a gene that encodes the alpha-1 antitrypsin, a serine protease inhibitor [75]. Interestingly, alpha-1 antitrypsin was demonstrated to counteract excitotoxicity in primary cortical neurons [76]. Using instead a candidate gene approach, Kim and colleagues focused on genes involved in DNA repair mechanisms. DNA damage accompanies aging, but also neurodegeneration and ALS [77]. Over 125 genes in the human genome encode proteins directly involved in DNA repair mechanisms [78]. The study validated the accumulation of DNA damage in post-mortem upper and lower MN from sALS and fALS patients, as well as the activation of DNA repair mechanisms. In addition, the study demonstrated hypomethylation of the promoters of the selected DNA repair genes *OGG1*, *APEX1*, *PNKP*, and *APTX* in patients compared to controls. The work showed that accumulation of DNA damage in vulnerable neurons does not arise from a failure of DNA repair mechanisms, but is rather actively addressed through the de-repression of relevant DNA repair genes [79]. In all, broad genome methylation analyses conducted on neural tissues and combined with gene ontology confirmed that neurodegeneration in ALS associates broadly with altered RNA and protein metabolism and neuroinflammation and involves active neuroprotection responses, revealing DNAm as a faithful witness of ongoing cellular events.

### 3.4. DNAm as an Epigenetic Disease Modifier

In addition to whole-genome methylation analyses, a particular focus has been given to the methylation of genes causally involved in ALS, both in sALS and fALS tissues. In the frontal cortex of sALS patients compared to controls, no difference was found in the methylation levels of *SOD1*, *TARDBP*, *VEGF*, or *ANG* [68,80]. Similarly, in the blood of *SOD1* mutation carriers *SOD1*, *TARDBP*, *FUS*, and *C9ORF72* were found demethylated, similarly to control individuals [81]. Upon blood analyses of monozygotic twins and triplets, Tarr and collaborators reported no difference in the methylation of *SOD1* in the disease discordant triplets that carried *SOD1* mutation, nor of *C9ORF72* in the disease discordant twins that carried a *C9ORF72* expansion [71]. However, larger cohorts used to interrogate the methylation level of *C9ORF72* promoter provided opposite results. As opposed to control individuals and non-*C9*-ALS patients, *C9ORF72* expansion carriers were reported to present with increased methylation of the *C9ORF72* promoter, both in the blood and in the frontal cortex [82,83], resulting in transcriptional repression [82,83,84]. The consequences included, at the molecular level, reduced intronic mRNA expression, reduced toxic RNA foci load, and decreased toxic dipeptide repeat protein (DPR) accumulation [84]. At the clinical level, increased *C9ORF72* methylation correlated with increased survival of *C9*-FTD patients [85] and neuroprotection of the frontal cortex and hippocampus in *C9ORF72* expansion carriers [86]. In a recent study conducted on the motor, occipital, and cerebellar cortices of sALS patients and controls, Koike and collaborators demonstrated that DNA demethylation in the auto-regulatory region of *TARDBP* 3′UTR reduces its alternative splicing, leading to increased *TARDBP* mRNA expression [87]. In addition, in the human motor cortex, they observed that demethylation of this region occurred with aging and correlated 3’UTR *TARDBP* demethylation with the age of disease onset [87]. Together, this recent work suggests that DNAm could possibly represent an epigenetic disease modifier, not necessarily at the level of the whole genome, but on relevant promoters and regulatory regions such as the *C9ORF72* repeat expansion in *C9*-ALS patients or the auto-regulatory element of *TARDBP* in the vast majority of sALS and fALS patients.

## 4. Histone Methylation

The second epigenetic layer concerns histone modifications. Histones are assembled in octamers of two H2A/H2B dimers and one H3/H4 tetramer, around which wraps 146 base pairs of DNA to compose the nucleosome [88]. The N-terminal tail of histones that protrudes outside the nucleosome is the subtract of a wide range of reversible modifications, the most studied being methylations and acetylations [89]. These are often found located at DNA-cis regulatory elements, including promoters and enhancers, and regulate the level of compaction of the chromatin and its accessibility by the transcriptional machinery, playing a central role in gene expression regulation [90,91]. Histone methylation consists of the addition of methyl group(s), from the methyl donor SAM, onto lysine (K) and arginine (R) residues, mostly of H3 and H4 histones [92]. Lysine residues can be mono-, di-, or tri-methylated. Lysine methylation is reversible and results from the activities of histone lysine methyltransferases (KMTs) and histone lysine demethylases (KDMs), which are numerous and highly specialized in given residues [92]. Arginine residues can be mono- or di-methylated, and their di-methylation can be either symmetric or asymmetric. Histone arginine methylation results from the activity of protein arginine methyltransferases (PRMTs). As of now, no specific arginine demethylase has been identified [92]. The combination of the nature and location of histone modifications makes up the so-called “histone code” that defines active, inactive, or poised/bivalent promoters. While H3K4me3 (and H3K27ac) typically mark active promoters, high levels of H3K9me2/3 (as well as DNAm) typically mark inactive promoters, and the co-occurrence of H3K4me3 and H3K27me3 characterizes instead bivalent promoters [91]. Importantly, post-translational histone modifications are tightly regulated across cell types and cell states and are crucial for proper neurogenesis, brain development, and CNS homeostasis. Mutations in genes encoding KMTs or KDMs cause neurodevelopmental disorders including intellectual disability, autism spectrum disorder, and Rett syndrome [93], altered post-translational histone modifications have been reported in neuropsychiatric disorders such as schizophrenia and depression, and a wide range of neurodegenerative diseases including ALS [92,94,95].

There has been so far a very limited number of studies designed to assess histone methylation in the context of ALS, and most of them have focused on the epigenetic consequences of known mutated gene expression, either in tissues obtained from fALS patients or in cellular or animal models. In addition, comparisons between the effects of different mutations on histone modifications are sparse. Therefore, it is at the moment difficult to identify histone-dependent epigenetic dysregulations that may be common to fALS or sALS cases, and a fortiori to all cases of ALS. Instead, comparisons between *SOD1*, *TARDBP*, and *FUS* mutation effects on the selected H3K4me2 and H3K14ac-S10ph (phospho-acetylation of serine 10 and lysine 14 on histone 3 tail) active marks and the H3K9me3 repressive mark in the cellular SH-SY5Y model revealed alterations that are gene- and even mutation-specific [96]. These findings are discussed in the next sections and summarized in Table 2.

### 4.1. Polycomb Repressive Complex 2 and Repressive H3K27me3 Mark

Polycomb repressive complex 2 (PRC2) catalyzes the tri-methylation of H3K27, leading to the repressive mark H3K27me3. Work in the field of HD demonstrated the neuroprotective functions of PRC2 that silences genes whose expression is detrimental to neuronal homeostasis and survival [97,98]. In post-mortem motor cortices from *C9*-ALS/FTD patients, PRC2 was found insoluble, suggesting a putative sequestration, possibly by G-quadruplex (G-Q) RNA structures formed by mutant *C9ORF72* mRNA [99]. While the consequences on H3K27me3 levels have not been reported, transcriptomic analyses conducted on post-mortem brains from *C9*-ALS/FTD patients have unraveled the de-repression of PRC2-target genes, suggesting that *C9*-mediated neurodegeneration likely involves epigenetic dysregulations, in addition to the production of toxic RNA and DPRs and loss of function [99,100,101] (Table 2). 

### 4.2. Lysine-Specific Demethylase 1 and the H3K4me2 Active Mark

Lysine-specific demethylase 1 (LSD1) demethylates the active mark H3K4me2, leading to transcriptional repression. Mutant *SOD1*-induced NSC34 cell degeneration as well as mouse *SOD1^G93A^* motoneuronal death are accompanied with increased expression of LSD1 protein and decreased H3K4me2 levels [102]. Knocking down *Lsd1* in mutant *SOD1*-expressing NSC34 cells was sufficient to increase neuronal survival [102]. In the *SOD1^G93A^* mouse model of ALS, treatment with polyamines, potent inhibitors of LSD1, rescued the LSD1-H3K4me2 pathway, prevented motoneuron degeneration, and improved motor performance and survival [102]. In line with a neuroprotective role of LSD1, *LSD1*-deficient mice were reported to present with neurodegeneration in the cerebral cortex and hippocampus, associated with paralysis and deficits in learning and memory [103]. Importantly, the transcriptional changes observed in *LSD1*-deficient mice overlap with those observed in AD and progranulin-associated FTD where LSD1 protein was found sequestered in phospho-Tau and phospho-TDP-43 pathologic inclusions, respectively [103]. In contrast, no overlap was found between the transcriptome of *LSD1*-deficient mice and that of ALS motoneurons, suggesting that altered LSD1-dependent transcriptomic regulations may be cortex/hippocampus-specific. Comparison with the recently published single-cell RNA sequencing data from the motor and prefrontal cortices of sporadic and *C9*-ALS patients and frontotemporal lobar degeneration (FTLD) patients would thus be particularly interesting [104].

### 4.3. Protein Arginine Methyltransferase 1 and the Active H4R3me2 Mark

Protein arginine methyltransferase 1 (PRMT1) is responsible for the asymmetric di-methylation of H4R3, leading to the active transcription mark H4R3me2, as well as the di-methylation of other proteins known as asymmetric di-methylated proteins (ASYM). Immunoreactivities of PRMT1 and ASYM were increased in the post-mortem spinal cord of sALS patients [105]. In addition, the concentration of asymmetric dimethyl l-arginine was higher in the cerebrospinal fluid of patients compared to controls, and the ratio of asymmetric dimethyl l-arginine/l-arginine correlated with disease severity [105]. Noteworthily, PRMT1 is mainly expressed during neural embryonic development, and its expression in adult CNS is triggered by stressors such as oxidative stress, hypoxia, and mitochondrial stress, which are typical hallmarks of neurodegeneration [105,106]. Thus, PRMT1 upregulation in sALS and its possible consequences on the epigenetic landscape and downstream transcriptome may occur rather late in the cascade of cellular events that mark neurodegeneration and could, therefore, represent a compensatory mechanism. In agreement with a neuroprotective role of *PRMT1* upregulation on sALS, conditional *PRMT1* knockout in motoneurons was recently demonstrated to trigger neurodegeneration and dismantlement of the neuromuscular junction leading to muscle wasting and lethality [107]. Importantly, this paradigm mimics PRMT1 loss of nuclear function, which accompanies *FUS* mutations. FUS is indeed a substrate of PRMT1, and PRMT1-mediated asymmetric arginine methylation of FUS is required for the proper nucleocytoplasmic shuttling of the protein [108,109,110,111,112]. FUS cytoplasmic mislocalization induced by *FUS* mutations triggers PRMT1 cytoplasmic retention leading to nuclear loss of function and decreased H4R3me2 levels, as reported in primary mouse motoneurons expressing mutant *FUS* as well as in motoneurons of the *FUS^R495X^* mouse line [108]. *C9ORF72*-ALS may instead involve PRMT1 gain-of-function mechanisms. *C9ORF72* repeat expansion mutations are responsible for the production of polyGR and polyPR DPRs, which, given their richness in arginine, are putative new substrates of PRMTs. Pharmacological inhibition of type I PRMTs (i.e., PRMT1, 2, 3, 4, 6, 8) was thus tested in cellular models and proved beneficial against polyGR/polyPR cytotoxicity [113], as well as against polyGR_15_-inducted toxicity in *C9* patient-derived MN [114]. 

### 4.4. Complex Patterns of Altered Histone Methylation Characterize C9ORF72 Pathology

Belzil and collaborators reported that reduced *C9ORF72* expression in the frontal cortex and cerebellum from *C9-*ALS and *C9*-FTD patients was associated with increased binding of *C9ORF72* to repressive histone methylation marks, namely H3K9me3, H3K27me3, H3K79me3, and H4K20me3 [115], providing a second mechanism for gene silencing in addition to the aforementioned DNA methylation and highlighting again the beneficial effect of epigenetic methylation reaction in silencing the toxic repeat expansions of *C9ORF72* [115]. A few years later, Esanov and collaborators not only confirmed increased H3K9me3 of *C9ORF72* in a humanized *C9*-BAC mouse model, but further demonstrated that this occurred as early as the first postnatal week of life, pointing to the efficiency of the neurodevelopmental period to set up first compensatory mechanisms [116]. In accordance with the idea of a compensatory mechanism, increased expression of the repressive methylated histone marks seems to selectively target the repeat expansions of *C9ORF72* as opposed to the whole genome, since global H3Kme3 levels were instead found reduced in astrocytes and neurons of the spinal, motor cortex, and hippocampus of the same *C9*-BAC mouse model and correlated with pathological and behavioral changes [117]. 

In a polyPR mouse model of ALS, Zhang and collaborators unraveled the colocalization of this toxic DPR with the repressive H3K9me3 mark and the active H3K4me3 mark and confirmed their findings in post-mortem cortices from *C9*-ALS/FTD patients [118]. These colocalizations were seen in heterochromatin, typically labeled by the repressive H3K9me3 mark but not the active H3K4me3 mark. Transcriptomic analyses further revealed altered gene expression that mostly corresponded to gene downregulation, with the notable exception of repetitive elements (REs), normally found silenced in heterochromatin, that were upregulated, leading to accumulation of double-stranded RNAs [118]. Global loss of heterochromatin is reminiscent of natural aging and, more particularly, in excitatory neurons [119]. In muscle stem cells, depletion of the methyl donor SAM was demonstrated to trigger loss of heterochromatin, linking 1C metabolism to histone methylation, chromatin composition, and aging [120], a connection that still awaits demonstration in neurons. 

Histone methylation studies in ALS are sparse and have been mostly conducted on genetic cases that have unraveled gene-specific alterations, along with great heterogeneity and complexity. More work is thus needed to understand whether common histone methylation impairments may emerge from the investigation of sALS cases. It is in addition important to note that cross-talks exist between the different types of epigenetic regulations, i.e., between DNAm and histone post-translation modifications, as well as among histone modifications. For instance, H4R3me2 is involved in the recruitment of histone acetyltransferases [121], which might explain the concomitant decrease in H4R3me2, H3K9ac, and H3K14ac levels in a yeast model of FUS proteinopathy, along with global decrease in gene transcription [122].

**Table 2 cells-13-00967-t002:** Reported altered histone methylation in ALS patients and models.

Tissue/Model	Observations	References
Post-mortem frontal cortex and cerebellar tissues of *C9*-ALS/FTD patients, sALS, and controls	Upregulation of PRC2-target genes in C9-ALS/FTD	[101]
Post-mortem brains of *C9*-ALS/FTD, ALSFTD non-expansion carriers, and controls	Upregulation of PRC2-target genes	[100]
Post-mortem motor cortex of *C9*-ALS/FTD patients and controls	Upregulation of PRC2-target genes Insolubility of the PRC2 catalytic subunit EZH2	[99]
MN from lumbar spinal cord of *SOD1^G93A^* mice	Increased expression of LSD1 Decreased H3K4me2 levels	[102]
NSC34 cells expressing *mSod1*	Increased expression of LSD1 Decreased H3K4me2 levels Increased neuronal survival upon LSD1 knockdown	
*SOD1^G93A^* mice	LSD1 inhibition through polyamine treatment Rescued LSD1-H3K4me2 pathway Improved MN survival and protected neuromuscular junctions Improved motor performance and survival	
MNs from spinal cord of *SOD1^G93A^* mice and SH-SY5HY cells expressing SOD1 mutants G93A and H80R	Slight reduction of H3K4me2 at presymptomatic and symptomatic stages	[96]
Post-mortem spinal cord of sALS patients and controls	Increased expression of PRMT1	[105]
Primary spinal cord cultures of mutant *FUS* MN	Mislocalization of PRMT1 in the cytoplasm with FUS Reduction in H4R3me2 and H3K9/K16 and transcriptional activity	[108]
Spinal cord of transgenic *FUS^R495^* mouse	Mislocalization PRMT1 in the cytoplasm together with FUS	
Post-mortem cortex of *C9*-FTD/ALS and GFP-(PR)_50_ mouse	Increased heterochromatin markers (H3K9me3 and H3K27me3) and euchromatin mark (H3K4me3) with nuclear polyPR colocalization in heterochromatin	[118]
Post-mortem frontal cortex and cerebellum of *C9*-ALS/FTD patients and controls	Increased H3K9me3, H3K27me3, H3K79me3, and H4K20me3 in the repeat expansion of *C9ORF72* gene	[115]
Cortex of the transgenic *C9*-BAC mice	Increased H3K9me3 in *C9ORF72* promoter and consequent reduction in its expression	[116]
Primary spinal cord astrocyte cultures and astrocytes and neurons of spinal cord, motor cortex, and hippocampus of transgenic *C9*-BAC mice	Reduced H3K9me3 nuclear levels	[117]

## 5. Pharmacological Targeting of Epigenetic Methylation in ALS

Different pharmacological candidates have already been tested in the context of ALS for their abilities to modulate DNA or histone methylation status (Table 3). As already mentioned, the pan type I PRMT inhibitor MS023 was reported to protect against polyPR and polyGR cytotoxicity in NSC34 cells [113] as well as in *C9* patient-derived MN [114]. Polyamines, which are potent LSD1 inhibitors, were found to protect against MN degeneration and improve motor performance and survival in the *SOD1^G93A^* mouse model of ALS [102]. The FDA-approved decitabine, a guanine/cytosine analog, which is incorporated into genomic DNA during replication, leading to sequestration and inactivation of DNMTs, was recently reported to reduce the expression of *C9ORF72*-derived toxic nuclear RNA foci and polyGA and polyGP DPRs in iPSC-derived cortical neurons from *C9ORF72* repeat expansion carriers, as well as in a BAC transgenic *C9* mouse model [123]. Yet, no global DNA methylation changes were detected, suggesting that the beneficial effect of decitabine was independent of DNMT inhibition [123]. In accordance with this, the two DNMT inhibitors 5-Fluoro-2′-deoxycitidine and RG108 did not show any effect on DPR production in this humanized culture model [123], in spite of the previously reported protective effect of RG108 in the *SOD1^G93A^* mouse model of ALS [57]. Decitabine efficiency was instead proposed to be mediated by impaired repeat transcription [123]. EPZ-6438 (Tazemetostat), an inhibitor of the enhancer of zeste homolog 2 (EZH2) responsible for H3K27 methylation, was found to reduce DNA damage in MN derived from a *C9*-ALS-iPSC line, but failed to correct TDP-43 subcellular mislocalization [124]. However, whether EPZ-6438 effect was mediated by the modulation of H3K27 methylation levels was not tested. BIX-01294, which inhibits the euchromatic histone-lysine N-methyltransferase 2 (RHMI2 or G9A), responsible for H3K9 methylation, was tested in an in vitro assay and reported to inhibit the repeat associated non-AUG (RAN) translation of the *C9ORF72* repeat expansion associated with ALS/FTD [125]. Yet, in cellular models, the compound increased polyGA expression and proved to be toxic [125] and, therefore, does not represent an interesting therapeutic candidate.

In addition to the aforementioned molecules, additional epidrugs exist that have been developed and are being tested in the field of cancer [126]. Given the high complexity and variability of the reported findings, it appears that drug screening efforts initiated by Czuppa and collaborators and Green and collaborators using patient-derived cellular models [123,125] should be pursued to identify repurposing candidates among these additional promising epidrugs.

## 6. Environmental and Developmental Impacts on 1C Metabolism

Given the contribution of vitamin B9 (folic acid) and vitamin B12, alongside the essential nutrient choline, 1C metabolism is highly dependent on diet [127,128,129] (Figure 1). More than 20% of adults above 60 years of age have a vitamin B12 deficiency, which is associated with increased risk of stroke, and could arise from insufficient dietary intake—in the case of a vegetarian diet—or decreased absorption [127]. Choline deficiency is associated with liver damage and/or muscle damage [128]. In a recent study conducted on a small cohort of sALS patients, prevalence of folate deficiency was considered severe and correlated with disease severity, along with other vitamin deficiencies [130]. In addition, excessive alcohol consumption, smoking, and ionizing radiations have also been associated with altered 1C metabolism or DNAm [131,132,133]. These may represent factors that, alone or in combination, and over time, may contribute to the emergence of ALS in individuals with initial genetic susceptibility. However, early and possibly transient 1C metabolism and methylation imbalance may also represent one of multiple early-life exposures expected to influence health and disease years later. DNMTs are indeed essential for proper nervous system development [134], and aberrant DNAm has been implicated in neurological disorders, including Rett syndrome [135]. Similarly, severely imbalanced prenatal 1C metabolism has neurodevelopmental consequences [136,137,138]. However, it remains to be evaluated whether milder DNAm, histone methylation, or 1C metabolism imbalance during development may have consequences that could at first remain silent and over time contribute to neurodegeneration onset. Dietary enrichment in folate and B12 vitamin has proven beneficial on disease progression in adult *SOD1^G93A^* mice [35]. The effects of such dietary supplementation and of the aforementioned epidrugs during development remain to be tested in mouse models of ALS, with a focus on those that carry mutations in genes proven to be required for proper neurodevelopment, such as *FUS*, *TARDBP*, and *C9ORF72* [139,140,141].

## Figures and Tables

**Figure 1 cells-13-00967-f001:**
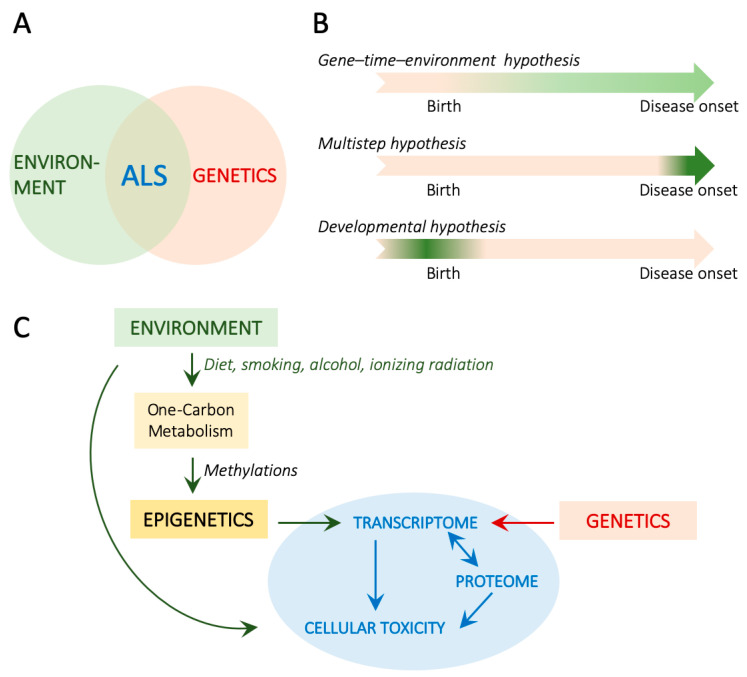
Genetic and environmental contributions to ALS. (**A**). Whether sporadic or familial, ALS arises from the combination of genetic susceptibility and environmental factors. (**B**). Different models are proposed to explain how genetic and environmental interactions may lead to disease onset. (**C**). Schematic representations of the molecular cascade leading to cellular toxicity in ALS.

**Figure 2 cells-13-00967-f002:**
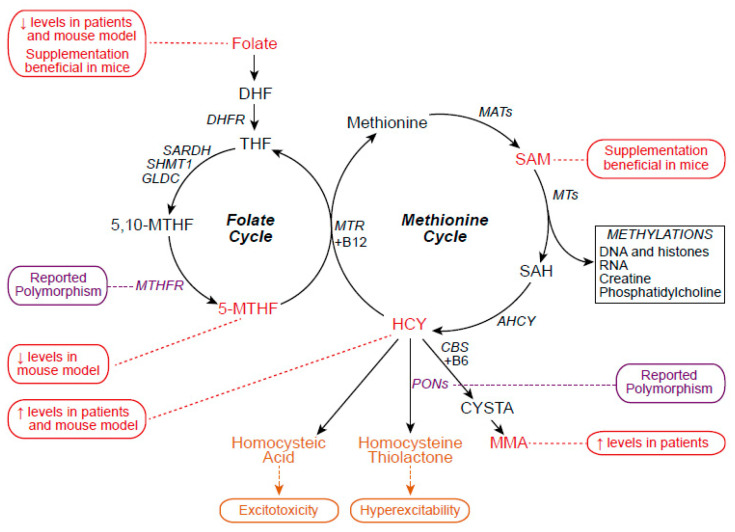
One-carbon metabolism and its impairments in ALS. ALS is associated with alterations of the folate and methionine cycles, such as reported polymorphisms of genes coding for key enzymes (purple), modified levels of metabolites (red), or altered neuronal functions that could indirectly arise from the production of toxic metabolites (orange).

**Table 1 cells-13-00967-t001:** Reported altered DNMT expression and DNA methylation in ALS patients and models.

Tissue/Model	Observations	References
Post-mortem motor cortex of sALS and controls	Increased DNMT1 and DNMT3A protein levels Accumulation of 5mC in pyramidal neurons	[49]
Post-mortem LMN fom sALS and controls	Increased DNMT1 and DNMT3A protein levels Accumulation of 5mC in LMN	
NSC34 cell line	DNMT3A overexpression induces apoptosis CPT-induced apoptosis is associated with increased DNMT3 expression Neuronal apoptosis is associated with 5mC accumulation Blocking DNMT3 activity or expression prevents neuronal apoptosis	
Post-mortem LMN from sALS and C9-ALS and controls	Accumulation of 5mC and 5hmC in LMN LMN with TDP-43-depleted nucleus had lower 5mC and 5hmC	[58]
Post-mortem upper and lower cortical neurons from C9-ALS patients and controls	Decreased DNMT3A expression	[60]
*FUS* iPSC-derived LMN	Increased *DNMT1*, *DNMT2*, and *DNMT3A* expression	[59]
Transgenic *hSOD1^G93A^* and *hSOD1^G37R^* mouse spinal cord and skeletal muscle	Increased DNMT1 and DNMT3 protein levels Increased DNMT activity Accumulation of 5mC in LMN and satellite cells of the skeletal muscle	[57]
Transgenic h*SOD1^G93A^*	Blocking DNMT3 activity delays disease onset, improves motor function, and extends survival	

**Table 3 cells-13-00967-t003:** Epidrugs targeting DNA and histone methylation in ALS.

Targets and Drugs	Effects	References
DNMT inhibition		
◾ Decitabine	Reduced DPR in *C9*-iPSC-derived neurons and *C9*-BAC mice, independent from DNMT inhibition	[123]
◾ RG108	Delayed disease progression in SOD1^G93A^ mice No effect reported in polyGP expression in *C9*-iPSC-derived cortical neurons and transgenic *C9* mice	[57] [123]
KMT inhibition		
◾ EPZ-6438 or Tazemetostat	Reduced DNA double-strand break in one *C9*-iPSC-derived MN culture without affecting TDP-43 subcellular mislocalization	[124]
◾ BIX-01294 (G9A inhibitor)	Binds repeat RNAs and inhibits RAN translation linked to *C9*-ALS/FTD but increases polyGA expression and induced cell death	[125]
PRMT inhibition		
◾ MS023 (type I PRMT inhibitor)	Protects against polyGR and polyPR toxicities in NSC-34 cells Protects against polyGR toxicity in *C9-*patient-derived MN	[113] [114]
KDM inhibition		
◾ Polyamines (LSD1 inhibitor)	Protect against MN degeneration, improve motor performance and survival in *SOD1^G93A^* mice	[102]

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
