# Peer review of "From Environment to Gene Expression: Epigenetic Methylations and One-Carbon Metabolism in Amyotrophic Lateral Sclerosis"

_cells, 2024, doi:10.3390/cells13110967_

Round 1

Reviewer 1 Report

Comments and Suggestions for Authors

There is currently a well-founded belief that it is from understanding epigenetic events that we will get answers that are still missing on the etiology of complex diseases. In the last decade an impressive scientific literature on epigenetics of neurodegenerative diseases has been formed, including the ALS. Marina Hernan-Godoy and Caroline Rouaux wrote a review of the literature on epigenetic alterations in the ALS that focuses on the methylation events reported in neural tissues,  as a possible consequence of alterations in metabolism 1C and epigenetic dysregulation in the ALS. The review is well done, it includes the main articles on this topic that the Authors have well put in relation to each other. The review is driven by an underlying hypothesis that the understanding the interplay between exposome, 1C metabolism, and epigenetic modifications will likely contribute to elucidating the mechanisms underlying altered gene expression related to ALS. This aspect makes this manuscript original compared to other recent literature reviews dealing with other aspects of the complicated relationship between epigenetics and ALS.

I have no major changes to suggest. Perhaps a greater synthesis on the description of the general aspects through which epigenetic mechanisms are implemented. But that’s just my opinion.

Author Response

We are thankful to the reviewer for the positive evaluation and the comment. We now provide a greater synthesis of the general aspects through which epigenetic mechanisms are implemented in the introduction, lines 69-94, along with other modifications highlighted in blue, throughout the manuscript.

Reviewer 2 Report

Comments and Suggestions for Authors

This review articel contains a very thorough, comprehensive and interesting overview about the epigenetic mechanisms that can be held responsible for the onset of the neurodegenrative disease ALS and for the pathological mechanisms underlying.

The paper is very well structured and understandable. As far as I can say mechansism of epigenetic regulation are well described in details reflecting the state of art scientific knowledge.  The comparison of mouse model (SOD, LSD1) results to the human disease depicted by the authors is very important, e.g., similarities and key differences in LSD deficient mice and in ALS motoneurons. 

I wonder whether based on their knowledge, the authors could try to draw conlcusions for and suggest candidates of pharmacological intervention or - in the case of fALS - prevention, or at least suggest the pharmacological properties that future drugs should provide in order to exert therapeutic effects. Is there the role of folic acid for treatment of prevention?

The final section about develomental impacts on ALS is quite interesting and therefore could be discussed more extensively. 

The table no. 2 is helpful with its content but the format is confusing and in sometimes not approporiately aligned in columns.

Author Response

We thank the reviewer for the positive evaluation and feedbacks.

We have added a new paragraph, lines 527-562 and new table (Table 3) to develop the pharmacological options.
The developmental impacts on ALS is now further detailed, lines 587-593.

We have realigned the text in Table 2.

All modifications are highlighted in blue throughout the manuscript.

Reviewer 3 Report

Comments and Suggestions for Authors

This work comprehensively outlines the current evidence on the interplay of epigenetics and cellular pathology in both familial and sporadic ALS. In particular, the authors put a focus on 1C metabolism, DNA methylation, and histone methylation. I strongly  commend  the authors for collecting and summarizing a huge amount of scientific evidence that impressively shows how important epigenetic factors are likely to be in ALS pathogenesis. However, I am afraid that the authors got bogged down in the multitude of details and lost sight of the big lines. On the one hand, this applies to a complete presentation of mechanisms relevant to epigenetics, all of which are mentioned in the introduction, but not elaborated on in the following parts of the manuscript. For example, the topic of miRNA expression and  RNA editing are not even roughly outlined.  On the other hand, although the authors postulate that knowledge of epigenetic influencing factors and mechanisms opens up the possibility of targeted therapies, they do not sufficiently address which pharmacological options could be considered as potentially promising. Since RNA-related epigenetics is not the focus of this review, RNA-based developments do not have to be covered here, but therapeutic approaches that target histone modifications should be addressed in order to build the bridge to the clinical application of mechanistic evidence. In particular, the HDAC inhibitor sodium phenylbutyrate (NaPB) should be mentioned. NaPB even reached approval in several countries based on an earlier phase II RCT, until recently a phase III RCT (PHOENIX) unfortunately did not prove efficacy.

Comments on the Quality of English Language

1. This article needs revision as there is a multitude of small linguistic errors, missing words, misused prepositions, and previously unintroduced abbreviations. Thorough language editing is strongly recommended in order to improve the overall readability of this manuscript.

2. The reference list contains "double-numbering" from the first to the last reference. Please revise.

Author Response

Thank you for these constructive comments.

We now have added in the introduction paragraphs related to histone acetylation and HDAC inhibitors, and miRNA, lines 69-90. In addition, to further highlight the focus if this review, we have slightly modified the title and the abstract.

We have added a new paragraph, lines 527-562 and new table (Table 3) to develop the pharmacological options.

We apologize for the inconvenience of the numerous small languages and editing errors throughout the text, that were due a final rush before submission. The text has now been thoroughly read and corrected. All changes are highlighted in blue. Finally, double numbering of the references, which was absent from our initially submitted word document have now been removed.

All modifications are highlighted in blue throughout the manuscript.

Reviewer 4 Report

Comments and Suggestions for Authors

 This review article by Godoy and Rouaux skillfully elucidates the intricate relationship among the exposome, 1C metabolism, and epigenetic modifications. It highlights how these interactions may unlock the mechanisms driving the alterations of gene expression associated with ALS.

While the review is well-composed, there are several suggestions for revision:

(1) The term 'epigenetics' in the title may be a little too broad. Limiting the focus on DNA/histone methylation would be more precise, as this is the main topic of the article;

(2) As indicated by the authors, the discussion on how the environmental impacts/factors-driven mechanisms influence the changes of DNA/ histone ALS could be expanded;

(3) The legend of Figure 2 refers to 'reported polymorphisms of genes, MTHFR and PONS'. This point could be clarified if there is evidence linking these polymorphisms to ALS.

Author Response

We are thankful to the reviewer for the positive evaluation comments, that we have addressed as follow.

(1) We agree with the reviewer comment and propose the modification highlighted in line 2.

(2) We now provide a new paragraph, lines 527-562 and new table (Table 3) to develop the pharmacological options, and have further developed the discussion on how environmental factors influence epigenetic changes in ALS, lines 587-593.

(3) Polymorphisms of MTHFR and PONs related to ALS have now been detailed in the text, lines 126-129.

All modifications are highlighted in blue throughout the manuscript.

Round 2

Reviewer 3 Report

Comments and Suggestions for Authors

The  authors have adequately revised  and amended the manuscript.